# CAffNet: Hard Constraint-Affine Neural Networks

**Yang Zhao**[1]  **Jungeun Lee**[2]  **Jeong hwan Jeon**[2]  **Sze Zheng Yong**[1]

## Abstract

We present a novel framework for embedding hard constraint satisfaction into neural network (NN) architectures, specifically feedforward neural networks and transformers, with input-dependent affine constraints of arbitrary cardinality. Traditional constraint enforcement approaches either rely on penalty-based soft constraints, which offer no guarantee of satisfaction, or on post-processing methods that enforce constraints after the NN is trained, which may lead to suboptimality. We introduce a trainable constraint-affine (CAffine) layer into NNs, yielding CAffNet, which goes beyond enforcing affine constraints via fixed orthogonal or parallel projections and enables joint optimization with network parameters. Moreover, we impose no restrictions on the constraint space dimensions and establish that our construction preserves the universal approximation properties of NNs, while providing provable guarantees on constraint adherence for all inputs. Experimental validation demonstrates robust performance across diverse domains requiring guaranteed constraint satisfaction.

## 1. Introduction

Neural networks have demonstrated remarkable capability in approximating complex nonlinear functions in high-dimensional spaces, making them attractive for a wide range of applications (Cybenko, 1989; Hornik et al., 1989). In safety-critical domains such as autonomous vehicles, robotics, and aerospace systems, this expressive power enables learned controllers that can adapt to complex dynamics and uncertainties (Kuutti et al., 2020; Brunke et al., 2022; Emami et al., 2022). However, ensuring that these learned controllers provide formal constraint satisfaction still remains a significant challenge (Brunke et al., 2022; Lavanakul et al., 2024).

A common approach to mitigate this issue is to incorporate soft constraints by adding a penalty term to the training objective, encouraging the network to produce safe outputs. While penalty-based methods can work well empirically, they generally do not provide hard safety guarantees, especially with limited training data or when encountering out-of-distribution edge cases at deployment (Márquez-Neila et al., 2017; Kolter & Manek, 2019; Min & Azizan, 2025).

In contrast, hard-constrained methods were studied by enforcing the network output within the safe region. Frerix et al. (2020) enforce homogeneous linear inequality constraints $Ax \leq 0$ by converting the feasible set from an H-representation to a V-representation of a polyhedral cone, and then restricting the network output to a combination of its rays via a constraint layer. However, this approach relies on the homogeneity of the constraints, and the hyperplane-to-vertex conversion can become computationally expensive as the output dimension and the number of constraints grow. In addition, C-DGMs (Stoian et al., 2024) add a differentiable constraints layer that incrementally adjusts each output element by computing the corresponding greatest lower bound and least upper bound, and clamping the value into this interval, working for a more general type of (affine) constraints $Ax \leq b$. Moreover, RAYEN (Tordesillas et al., 2023) can satisfy different kinds of constraints, such as linear, quadratic, second-order cone, and linear matrix inequality, by adjusting the output vector length from the interior point of the feasible region. However, these methods are only applicable when the constraints are input-independent.

Moving beyond input-independent constraints, recent work has begun to study hard constraint enforcement when the feasible set varies with the input (cf. Table 1). For input-dependent equality constraints, POLICE (Balestriero & LeCun, 2023) provably enforces the output to be an affine function of the input within a convex region by modifying layer parameters based on its vertices, while KKT-hPINN (Chen et al., 2024) enforces hard linear equality constraints by inserting projection layers derived from KKT conditions, and ACnet (Beucler et al., 2021) reformulates input-output nonlinear equality constraints to affine equality constraints.

---

[1]Department of Mechanical and Industrial Engineering, Northeastern University, Boston, MA 02115 USA [2]Department of Electrical Engineering, Ulsan National Institute of Science and Technology, Ulsan 44919, Republic of Korea. Correspondence to: Sze Zheng Yong <s.yong@northeastern.edu>.

*Proceedings of the $43^{rd}$ International Conference on Machine Learning*, Seoul, South Korea. PMLR 306, 2026. Copyright 2026 by the author(s).

*Table 1.* Comparison of methods enforcing input-dependent constraints on neural networks for the target function $y = f(x) \in \mathbb{R}^{n_{\text{out}}}$, based on the more detailed comparison in (Min & Azizan, 2025, Table 1). Our proposed CAffNet guarantees hard constraint satisfaction with a closed-form framework that preserves the universal approximation property for an arbitrary number of input-dependent affine inequality constraints[*].

| Method | Constraint | Support Eq./Ineq. | Satisfaction Guarantee | Computation | Universal Approx. |
|---|---|---|---|---|---|
| Soft-Constrained | Any | Both | No | Closed-Form | Yes |
| POLICE (Balestriero & LeCun, 2023) | $y = Ax + b \ \forall x \in R$ | Eq. Only | Always | Closed-Form | Unknown |
| KKT-hPINN (Chen et al., 2024) | $Ax + By = b$ (# constraints $\leq n_{\text{out}}$) | Eq. Only | Always | Closed-Form | Unknown |
| ACnet (Beucler et al., 2021) | $h(x, y) = 0$ (# constraints $\leq n_{\text{out}}$) | Eq. Only | Always | Closed-Form | Unknown |
| DC3 (Donti et al., 2021) | $g(x, y) \leq 0, h(x, y) = 0$ | Both | Asymptotic | Iterative | Unknown |
| FSNet (Nguyen & Donti, 2025) | $g(x, y) \leq 0, h(x, y) = 0$ | Both | Asymptotic | Iterative | Unknown |
| HardNet-Aff (Min & Azizan, 2025) | $b^l(x) \leq A(x)y \leq b^u(x)$ (# constraints $\leq 2n_{\text{out}}$) | Both | Always | Closed-Form | Yes |
| CAffNet (Ours) | $A(x)y \leq b(x)$ (arbitrary # constraints) | Both | Always | Closed-Form | Yes |

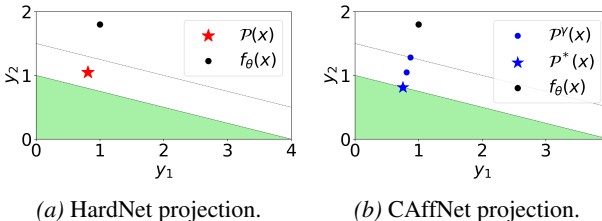

*(a)* HardNet projection.    *(b)* CAffNet projection.

*Figure 1.* Comparison of the projection results between HardNet (Min & Azizan, 2025) and the proposed CAffNet in the presence of linearly dependent constraints, where HardNet projects to a point outside the feasible region while CAffNet finds a feasible projection.

On the other hand, for the more general case with input-dependent inequality constraints, DC3 (Donti et al., 2021) uses a differentiable procedure to satisfy equality constraints and unrolls gradient-based corrections for inequality constraints. However, the approach relies on iterative optimization, whose convergence and runtime can vary across problem instances and hyperparameter settings. A similar problem also arises in FSNet (Nguyen & Donti, 2025), which solves an unconstrained optimization problem to minimize constraint violations iteratively. While this iterative procedure asymptotically drives violations toward zero, exact feasibility generally depends on running a sufficient number of iterations and meeting a tolerance.

Furthermore, projection-based methods onto a single (hard) constraint with closed-form expressions have been studied (Kolter & Manek, 2019; Min et al., 2023). It was then extended to multiple constraints by HardNet-Aff (Min & Azizan, 2025), which projects the output of neural networks into the safe region of input-dependent affine constraints by adding correction terms to the output. However, it assumes that $A(x)$ has full row rank. This requirement leads to two main limitations. One is that the number of (two-sided) constraints cannot exceed the input dimensions, which is

restrictive in real-world applications. Another limitation is that it cannot be applied to linearly dependent constraints (cf. Fig. 1a), where the projection may fall outside the feasible region. In contrast, our proposed CAffNet method addresses this issue as shown in Fig. 1b. Moreover, when $A(x)$ is not full column rank, the projection solution may not be unique, which is not considered in HardNet. For example, $\{(x, y) \in \mathbb{R}^2 \mid [0, 1][x, y]^\top \leq [0, 0]^\top\}$ has infinitely many solutions as the line of $y = 0$ satisfies the constraints. While HardNet only projects onto a specific point on this line, our proposed CAffNet introduces an additional trainable term to find the optimal projection onto this entire line instead of simply using orthogonal or parallel projection. A comparison of related input-dependent methods is summarized in Table 1. More comparisons with input-independent methods can be found in (Min & Azizan, 2025, Table 1).

Noting the above limitations, we propose CAffNet, which combines a constraint-affine (CAffine) layer with neural networks, to address these issues by introducing a constraint decomposition method and a trainable projection module to handle an arbitrary number of constraints with provable feasibility guarantees and universal approximation property.

The main contributions of this paper are as follows:
1. **Closed-form hard constraint enforcement.** We propose CAffNet, a closed-form neural network framework that guarantees hard satisfaction of an arbitrary number of input-dependent affine inequality constraints[*] without full row rank assumptions.
2. **Robust projection via constraint decomposition.** A constraint decomposition with a trainable projection module is proposed to handle linearly dependent constraints and non-unique projections.
3. **Architectural versatility and superior empirical performance.** CAffNet can be instantiated with feedforward

---
[*] Note that equality constraints can be equivalently expressed using two sets of inequalities with $\leq$ and $\geq$.

and transformer architectures, achieving zero constraint violations and reducing the MSE by 73.33% compared to the soft-constrained NN.

**Conflict of Interest Disclosure.** The authors declare no financial conflicts of interest or other substantive conflicts that could reasonably be perceived to influence this work.

## 2. Preliminaries

### 2.1. Notation

We denote by $\|x\|_p$ the $p$-norm for a vector $x \in \mathbb{R}^n$, and by $\|A\|_p$ the induced matrix norm for a matrix $A \in \mathbb{R}^{m \times n}$, where $p \in [1, \infty)$. We use $\mathcal{C}(\mathcal{X}, \mathcal{Y})$ to denote the space of continuous functions from $\mathcal{X}$ to $\mathcal{Y}$ endowed with the supremum norm $\|f\|_{\mathcal{C}} := \sup_{x \in \mathcal{X}} \|f(x)\|_\infty$, and $L^p(\mathcal{X}, \mathcal{Y})$ to denote the class of $L^p$ functions from $\mathcal{X}$ to $\mathcal{Y}$ with the $L^p$ norm $\|f\|_{L^p} := \left( \int_{\mathcal{X}} \|f(x)\|_p^p dx \right)^{1/p}$. For a matrix $A$, we use $A^\dagger$ to denote its Moore-Penrose pseudoinverse. In addition, we use $I$ to denote the identity matrix of appropriate dimension, and $\mathbb{1}_n$ to denote the vector of all ones in $\mathbb{R}^n$. Moreover, we use $\mathrm{diag}(\cdot)$ to denote a diagonal matrix.

### 2.2. Universal Approximation Theorem

The Universal Approximation Theorem (UAT) is fundamental to understanding the theoretical limits of neural networks for function approximation. It guarantees that, given sufficient width or depth, a neural network can approximate any continuous function to arbitrary accuracy.

**Definition 2.1** (Universal Approximator). For function classes $\mathcal{F}_1, \mathcal{F}_2 \subset \mathcal{C}(\mathcal{X}, \mathcal{Y})$ (respectively, $\mathcal{F}_1, \mathcal{F}_2 \subset L^p(\mathcal{X}, \mathcal{Y})$), we say $\mathcal{F}_1$ universally approximates (or is dense in) $\mathcal{F}_2$ if for any $f_2 \in \mathcal{F}_2$ and $\epsilon > 0$, there exists $f_1 \in \mathcal{F}_1$ such that $\|f_2 - f_1\|_{\mathcal{C}} \leq \epsilon$ (or more generally by norm equivalence, $\|f_2 - f_1\|_{L^p} \leq \epsilon$).

This property is well-established for unconstrained feedforward neural networks and transformers. For instance, a deep feedforward neural network with ReLU activations can universally approximate any $L^p$ function with width $w \geq \max(n_{in} + 1, n_{out})$ (Park et al., 2021) while a transformer can universally approximate any $L^p$ function with 2 heads of size 1 and a hidden layer size of 4 (Yun et al., 2019).

While universal approximation theorems establish that neural networks can approximate arbitrary continuous functions to any desired accuracy, they provide no guarantees regarding constraint satisfaction. Traditional constraint enforcement approaches either rely on penalty-based soft constraints, which provide no assurance of satisfaction, or postprocessing methods that enforce constraints after the neural network is trained, which may lead to sub-optimality. Therefore, developing principled approaches that simultaneously

guarantee approximation accuracy and provable constraint satisfaction represents a fundamental open challenge for their deployment in practical applications.

## 3. Methods

In this section, we propose CAffNet, a framework, shown in Fig. 2, that combines a constraint-affine (CAffine) layer with neural networks (NNs) to strictly enforce input-dependent affine constraints[*] on the output $y = \mathcal{P}^*(x) \in \mathbb{R}^{n_{out}}$, i.e.,

$$A(x)y \leq b(x), \tag{1}$$

where $x \in \mathbb{R}^{n_{in}}$ denotes the input, $n_{in}$, $n_{out}$ are input and output dimensions, respectively, $A(x) \in \mathbb{R}^{m \times n_{out}}$, $b(x) \in \mathbb{R}^m$, and $m$ is the number of constraints, with no restrictions on $m$ nor the rank of $A(x)$. The framework processes the input $x$ through two NNs to generate an unconstrained $f_\theta(x)$ and a null-space component $w_\phi(x)$. The NNs can be parameterized by any trainable universal function approximator establishing a mapping from input to output, including standard architectures like feedforward NNs or transformers, e.g., (Park et al., 2021) or (Yun et al., 2019). To facilitate any number of input-dependent affine constraints, the applicable constraints are decomposed into sub-constraints, which will be introduced in Section 3.1. Then, in Section 3.2, we present the closed-form projection architecture that integrates the unconstrained output, the null-space component, and the sub-constraints to guarantee hard constraint satisfaction, feasibility, as well as the universal approximation property.

### 3.1. Constraint Decomposition

To enforce an arbitrary set of linear/affine inequalities with provable feasibility guarantees, we decompose the constraint set into smaller subsets that can be handled by closed-form projection modules. This is inspired by the fact that the boundary of a polyhedron is determined by a smaller number of active constraints. It is reasonable to assume that the set $\mathcal{S}(x) := \{y \in \mathbb{R}^{n_{out}} \mid A(x)y \leq b(x)\}$ defined by the constraints in (1) is feasible. Since the feasible affine constraints form a polyhedron, we notice that it is not necessary to consider all sub-constraints, given the definition of minimal faces of a polyhedron below (Schrijver, 1998, Theorem 8.4):

**Definition 3.1.** (Minimal Face of Polyhedron) A set $F$ is a minimal face of a polyhedron $P = \{y : Ay \leq b\}$, if and only if $F \subseteq P$, $F \neq \emptyset$, and $F = \{y \mid A'y = b'\}$ for some sub-constraint $A'y \leq b'$ of $Ay \leq b$.

From the definition above, the minimal face is a polyhedron that does not contain any other faces of this polyhedron. In addition, all points on the minimal faces satisfy the original affine constraints. For any non-empty polyhedron, it needs

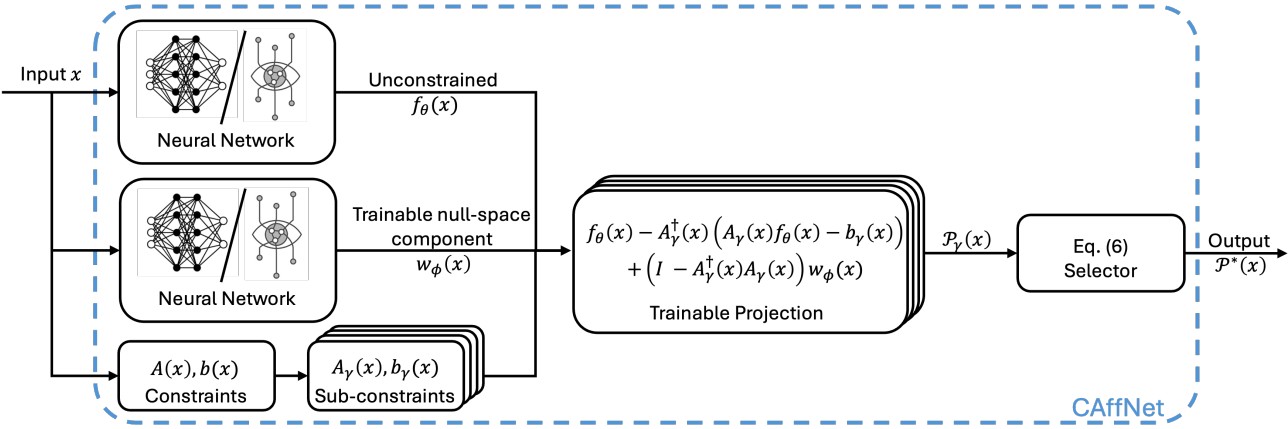

*Figure 2.* CAffNet Framework. Sub-constraints and a trainable null-space component are considered to find the optimal output, which is guaranteed to be within the safe region defined by input-dependent affine constraints.

at most $n_{out}$ linearly independent constraints to define a minimal face, which includes the vertices. Therefore, if the number of constraints $m$ exceeds the output dimension $n_{out}$, we only need to consider at most $n_{out}$ constraints. Otherwise, when $m \leq n_{out}$, the minimal faces are hyperplanes that can be at most defined by all $m$ constraints. Based on this observation, let integer $k$ represent the number of constraints chosen from $A(x)$ and $b(x)$ to form the sub-constraints, and $1 \leq k \leq \min(m, n_{out})$. Further, let integer $i$ denote the $i$-th constraint among the selected $k$ constraints, and integer $j_i$ denote the corresponding constraint index in the original $m$ constraints of $A(x)$, $b(x)$. Then, we have $1 \leq i \leq k$ and $1 \leq j_i \leq m$, we let $\gamma = (j_1, \ldots, j_i, \ldots, j_k)$ denote the index sequence. Considering all possible combinations of $k$ constraints, the set of corresponding index sequences is given by

$$\Gamma_k := \big\{ \gamma = (j_1, \ldots, j_i, \ldots, j_k) \mid \\ 1 \leq j_1 < \cdots < j_i < \cdots < j_k \leq m \big\},$$

and the set of all such combinations with $k \in [1, \min(m, n_{out})]$ is given by $\Gamma = \bigcup_{k=1}^{\min(m, n_{out})} \Gamma_k$, resulting in a *finite* total number of combinations:

$$\sum_{k=1}^{\min(m, n_{out})} \binom{m}{k} \leq 2^m - 1. \tag{2}$$

For each index sequence $\gamma = (j_1, \ldots, j_k) \in \Gamma$, we define the corresponding submatrix of $A(x)$ and subvector of $b(x)$ with rows $a_{j_i}^\top$ and $b_{j_i}$ selected by $\gamma$ as

$$A_\gamma = \begin{bmatrix} a_{j_1} & \ldots & a_{j_i} & \ldots & a_{j_k} \end{bmatrix}^\top \in \mathbb{R}^{k \times n_{out}}, \\ b_\gamma = \begin{bmatrix} b_{j_1} & \ldots & b_{j_i} & \ldots & b_{j_k} \end{bmatrix}^\top \in \mathbb{R}^k. \tag{3}$$

Then, the original constraints can be decomposed into multiple sub-constraints defined by $A_\gamma$ and $b_\gamma$ for all $\gamma \in \Gamma$, and each sub-constraint has a number of constraints not exceeding the dimension of $y$, i.e., $k \leq \min(m, n_{out}) \leq n_{out}$.

### 3.2. Projection with Sub-Constraints

We will now present the projection architecture that guarantees hard satisfaction of the input-dependent affine constraints based on the decomposed sub-constraints. We first introduce the following assumption on the continuity and feasibility of the constraints.

**Assumption 3.2.** For any input $x \in \mathbb{R}^{n_{in}}$, $A(x)$ and $b(x)$ are continuous in $x$, and the feasible region $\mathcal{S}(x) := \{y \in \mathbb{R}^{n_{out}} \mid A(x)y \leq b(x)\}$ is non-empty.

Using the decomposition method in Section 3.1 to obtain sub-constraints $(A_\gamma(x), b_\gamma(x))$, we derive the projection based on the solution set of the equations $A_\gamma(x)y = b_\gamma(x)$. If $A_\gamma(x)$ has full column rank, this equation has a unique solution; otherwise, there is a null space that makes the solution non-unique. This intuition leads to the following projection onto the sub-constraint $(A_\gamma(x), b_\gamma(x))$:

$$\mathcal{P}_\gamma(x) = f_\theta(x) - A_\gamma^\dagger(x)(A_\gamma(x)f_\theta(x) - b_\gamma(x)) \\ + (I - A_\gamma^\dagger(x)A_\gamma(x))w_\phi(x), \tag{4}$$

where $A_\gamma^\dagger(x)$ is the pseudoinverse of $A_\gamma(x)$, $f_\theta(x) \in \mathbb{R}^{n_{out}}$ is the output of an unconstrained NN, and $w_\phi(x) \in \mathbb{R}^{n_{out}}$ is a learned vector (also parameterized by an NN) representing the null-space component. The projected output has the following property, whose proof is provided in Appendix A:

**Lemma 3.3** (Existence of a Feasible Solution). *For any input $x \in \mathbb{R}^{n_{in}}$, if the feasible region $\mathcal{S}(x) := \{y \in \mathbb{R}^{n_{out}} \mid A(x)y \leq b(x)\}$ is non-empty, then there exists at least one candidate $y \in \{\mathcal{P}_\gamma(x) \mid \gamma \in \Gamma\}$ such that $A(x)y \leq b(x)$.*

Hence, the feasible candidate set can be defined by selecting all projections that are inside the safe region, i.e.,

$$\mathcal{S}_\mathcal{P}(x) := \{\mathcal{P}_\gamma(x) \mid \gamma \in \Gamma, \ A(x)\mathcal{P}_\gamma(x) \leq b(x)\}. \tag{5}$$

Finally, the output of CAffNet is defined as either the original output of the unconstrained NN, $f_\theta$, if it satisfies the

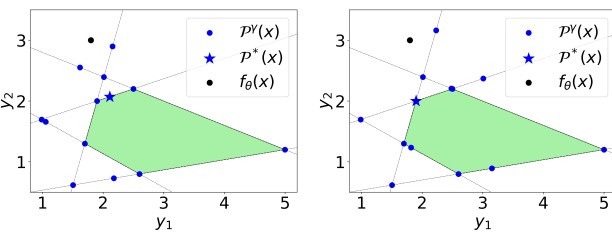

*(a) Projection onto an edge.*   *(b) Projection onto a vertex.*

*Figure 3.* Optimal projection results of CAffNet for different cases. The optimal final point may be different for various $w_\phi(x)$.

constraints, or the projection onto (5) that has minimum distance to $f_\theta$ (with any $p$-norm), i.e.,

$$\mathcal{P}^*(x) = \begin{cases} f_\theta(x), & \text{if } A(x)f_\theta(x) \leq b(x), \\ \arg\min_{y \in \mathcal{S}_\mathcal{P}(x)} \|y - f_\theta(x)\|_p, & \text{otherwise.} \end{cases} \quad (6)$$

Based on the construction above, CAffNet is guaranteed to satisfy the input-dependent affine constraints, as stated by the following theorem that is proven in Appendix B:

**Theorem 3.4** (Hard Constraint Satisfaction). *For any input $x \in \mathbb{R}^{n_{in}}$, under Assumption 3.2, the output of CAffNet $\mathcal{P}^*(x)$ in (6) satisfies the input-dependent affine constraints, i.e., $A(x)\mathcal{P}^*(x) \leq b(x)$.*

Note that CAffNet does not require $A(x)$ to be full-rank or irredundant. In particular, it does not restrict the number of constraints $m$ relative to the output dimension $n_{out}$, and it remains valid when constraints are redundant or linearly dependent. When a chosen submatrix $A_\gamma(x)$ has full column rank, i.e., $rank(A_\gamma) = n_{out}$ and hence $k = m$, we have $I - A_\gamma^\dagger A_\gamma = 0$. So, the null-space component disappears, and the projection is uniquely determined by the intersection of the selected constraints, i.e., an extreme point of the polyhedron. When $A_\gamma(x)$ is not full column rank, the constraint equations admit infinitely many solutions forming an affine subspace (a face/edge/hyperplane). Then, the null-space component $w_\phi(x)$ can be trained to select an optimal point within the null space, allowing different feasible points on the same face to be selected based on the task objective. If $w_\phi(x)$ is zero, it performs an orthogonal projection. The effect of this null-space term is illustrated in Fig. 3, where the optimal feasible point can lie on a vertex or an edge for different $w_\phi(x)$'s. Further, HardNet is recovered as a special case. If $m = n_{out}$, $A(x)$ has full row rank, and $f_\theta(x)$ violates all constraints, there exists a choice $\gamma$ for which $A_\gamma(x) = A(x)$ and $I - A_\gamma^\dagger A_\gamma = 0$. In this case, CAffNet's null space vanishes, and the resulting projection coincides with HardNet. Thus, CAffNet generalizes HardNet.

In addition to the feasibility and constraint satisfaction guarantees, CAffNet is also a universal approximator:

**Theorem 3.5** (Universal Approximation of CAffNet). *Suppose $\mathcal{X} \subset \mathbb{R}^{n_{in}}$ is compact, $A(x)$ and $b(x)$ are continuous over $\mathcal{X}$, and $\mathcal{S}(x) := \{y \in \mathbb{R}^{n_{out}} \mid A(x)y \leq b(x)\}$ is a non-empty set. Then, for any function classes $\mathcal{F}_\theta, \mathcal{F} \subset \mathcal{C}(\mathcal{X}, \mathbb{R}^{n_{out}})$ (or $\mathcal{F}_\theta, \mathcal{F} \subset L^p(\mathcal{X}, \mathbb{R}^{n_{out}})$ for any $p \in [1, \infty)$), if the unconstrained $\mathcal{F}_\theta$ universally approximates $\mathcal{F}$, then $\mathcal{F}_{CAffNet} := \{\mathcal{P}^*(x)\}$ universally approximates $\mathcal{F}_{target} := \{f_t \in \mathcal{F} \mid A(x)f_t \leq b(x)\}$.*

The proof is provided in Appendix C. As a corollary, since feedforward NNs and transformers can universally approximate arbitrary continuous functions (Park et al., 2021; Yun et al., 2019), CAffNet with the same feedforward NNs and transformers can also universally approximate the constrained target function.

*Remark* 3.6 (Constraint-Specific Null-Space Component). To enable more flexible and expressive learning of projections onto the appropriate hyperplanes, we could learn one null-space component for each sub-constraint, i.e., $w_{\phi,\gamma}(x)$. However, this would require a separate network for each sub-constraint, which can be computationally expensive and memory intensive. Therefore, we use a single shared null-space component $w_\phi(x)$ across all sub-constraints. This choice is more efficient in terms of both computation and memory, and as shown in Appendix C, is sufficient to preserve the universal approximation property.

*Remark* 3.7 (Scalability). We now discuss some potential scalability concerns of CAffNet and how to mitigate these issues in practice.

1. CAffNet-Lite: Choosing constraint combinations with cardinalities from 1 to $\min(m, n_{out})$ sometimes may be computationally expensive and memory intensive, especially when the number of constraints $m$ and the output dimension $n_{out}$ are large. To reduce the number of sub-constraints, we use the observation that the feasibility in Lemma 3.3 and its proof in Appendix A is certified by considering minimal faces, such as vertices for the pointed polyhedron. Therefore, in practice, one may restrict the candidate set to a subset of combinations, for example by considering each individual constraint and the combinations corresponding to minimal faces. In this case, $k \in \{1, \min(m, n_{out})\}$, and the total number of combinations is reduced to $m + \binom{m}{\min(m,n_{out})}$, which is polynomial in $m$, specifically $O(m^{\min(m,n_{out})})$, compared with the exponential complexity $O(2^m)$ of the full combination set. We refer to this reduced variant as CAffNet-Lite, which provides a practical way to apply the proposed method to larger-scale problems and reduce training time. Detailed derivation and discussion of this lite version are provided in (Zhao et al., 2026).

2. Complexity of Pseudoinverse Computation: Computing pseudoinverses can be expensive for large matrices. In our approach, the largest matrix requiring a pseudoinverse has

size $\min(m, n_{out}) \times n_{out}$, which is often manageable, and pseudoinverse computation is highly optimized in modern libraries such as PyTorch/CUDA (Paszke et al., 2019) through batched operations. For problems where the training data are available in advance, such as in Experiments 4.1 and 4.2, both $A_\gamma(x)$ and $A_\gamma^\dagger(x)$ can be precomputed for each data point and stored in a lookup table, which can then be reused during training and inference to reduce computation. The projection module can also benefit from GPU acceleration through parallelization and batch processing.

3. Hardware Limitations: For large-scale problems, the number of constraint combinations may be limited by hardware resources such as GPU memory. In such cases, the combinations can be processed in batches to accommodate hardware limitations.

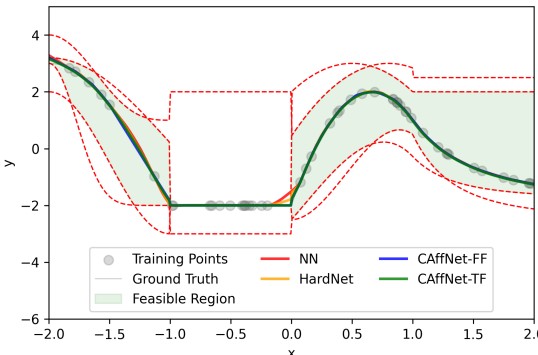

*Figure 4.* Learned functions with piecewise constraints. Both NN with soft constraints and HardNet have constraint violations around $x = -1$ and $x = 0$, while both CAffNets satisfy all constraints.

# 4. Experiments

In this section, we demonstrate the hard constraint satisfaction guarantee and effectiveness of CAffNet with three implementation scenarios, following similar experiments to (Min & Azizan, 2025) for learning (1) nonlinear functions with piecewise constraints, (2) optimization solvers, and (3) control policies in safety-critical systems. Since HardNet (Min & Azizan, 2025) has been shown to often outperform all other approaches, for simplicity, we only compare our proposed CAffNet with HardNet as well as standard neural networks (NNs) with soft constraints, using a penalty cost of $100\text{ReLU}(A(x)y - b(x))$, where $y$ is the output of each method. In all cases, the number of constraints is set to exceed the output dimension to demonstrate the advantage of CAffNet in handling such cases, and hence, soft constraints are also applied to HardNet, as it can have constraint violations. In addition, to demonstrate the flexibility of the CAffNet framework, we apply the proposed framework using both a feedforward neural network (CAffNet-FF) and a transformer-based model (CAffNet-TF). Each CAffNet-FF model uses ReLU activation functions and consists of three hidden layers with 200 neurons per layer, as in (Min & Azizan, 2025), while for CAffNet-TF, the transformer architecture is composed of three attention heads of size 40 and a hidden layer size of 120, chosen to have a comparable number of resulting parameters/decision variables as that of CAffNet-FF. For all experiments, we use the Adam optimizer with a learning rate of 0.0001 and the 2-norm in (6). Each experiment is repeated five times with different random seeds. Other detailed parameters for each experiment are described in the corresponding subsections. All models are implemented and trained using PyTorch (Paszke et al., 2019). The training and testing are performed with NVIDIA Tesla V100-SXM2-32GB.

The evaluation metrics include total loss, constraint violations, and computation times for training and testing/inference. The total loss is computed in different forms for each experiment and averaged over all training samples. The constraint violation is evaluated by $r = \text{ReLU}(A(x)y - b(x))$, where $r \in \mathbb{R}^m$. We report the maximum value of $r$, the mean value of $r$, and the percentage of violated constraints over all testing samples. The training time $T_{train}$ and testing time $T_{test}$ are measured in milliseconds (ms), where $T_{train}$ is the average training time for each training epoch, and $T_{test}$ is the inference time for the test set. Finally, the values without parentheses in the tables are the mean values over five runs, and the values within parentheses are the corresponding standard deviations.

## 4.1. Learning Nonlinear Functions with Piecewise Constraints

We first consider the problem of learning a nonlinear function with piecewise constraints. The target function is a piecewise nonlinear function $f : [-2, 2] \rightarrow \mathbb{R}$, with $n_{in} = n_{out} = 1$. There are four piecewise constraints; two are treated as upper bounds, and the other two are treated as lower bounds. The detailed description of the target function and the constraints are provided in Appendix D.1. We randomly generate 50 training samples on the target function with uniform distribution, and 400 testing samples linearly spaced in $[-2, 2]$. The training loss is the mean squared error (MSE) between the network output and the target function value. The training is performed for 50000 epochs with a batch size of 500. The learned functions are shown in Fig. 4, and the evaluation results are summarized in Table 2. It can be observed that both CAffNet variants strictly satisfy all constraints, while the others have constraint violations. Due to the more complex framework, both CAffNet variants take slightly longer to train and test compared to the others. However, CAffNet improves the training efficiency, with a significantly lower initial loss and a faster convergence rate as shown in Fig. 5, and it can also be observed that CAffNet-TF achieves the lowest MSE among all methods.

*Table 2.* Comparison of learning methods for nonlinear functions with piecewise constraints (mean; standard deviation in parentheses). Both CAffNets strictly satisfy all constraints with CAffNet-TF achieving the lowest MSE, while NN and HardNet have constraint violations. Compared to NN, CAffNet-TF reduces MSE by 73.33%.

| METHOD | MSE | INEQUALITY VIOLATION | | $T_{train}$ (MS) | $T_{test}$ (MS) |
|---|---|---|---|---|---|
| | | MAX | MEAN | | |
| NN | 0.0045 | 0.0074 | 0.0019 | **5.3100** | **3.3700** |
| | (0.0057) | (0.0074) | (0.0019) | (0.16) | (0.20) |
| HARDNET | 0.0037 | 0.0033 | 0.0008 | 8.0000 | 3.7000 |
| | (0.0063) | (0.0037) | (0.0009) | (0.34) | (0.24) |
| CAFFNET-FF | 0.0020 | **0.0000** | **0.0000** | 11.9300 | 7.1100 |
| | (0.0032) | (0.0000) | (0.0000) | (0.55) | (0.49) |
| CAFFNET-TF | **0.0012** | **0.0000** | **0.0000** | 15.5700 | 14.7100 |
| | (0.0009) | (0.0000) | (0.0000) | (0.32) | (2.04) |

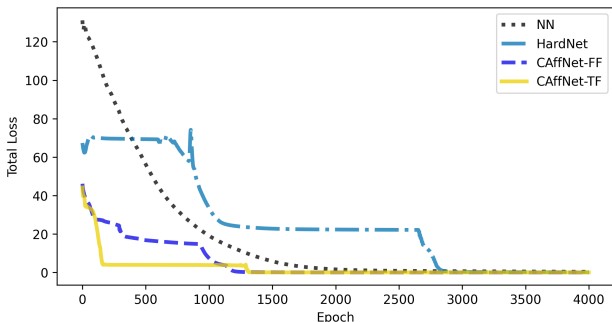

*Figure 5.* Training loss of the piecewise constraints experiment for the first 4000 epochs. Both CAffNet variants have a lower initial loss and a faster convergence rate than the rest.

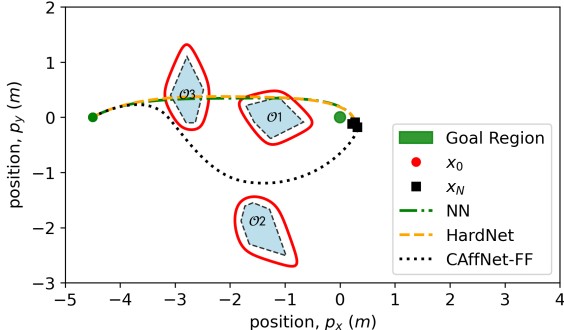

*Figure 6.* Trajectories of learning control policies for safety-critical systems. Both CAffNet-FF and HardNet arrive in the vicinity of the goal region. However, only CAffNet-FF strictly satisfies all safety constraints during the entire trajectory, while HardNet collides with the obstacle. NN with soft constraints also collides with the obstacle and goes far away from the goal region.

## 4.2. Learning Optimization Solvers

Next, we consider the problem of learning optimization solvers with inequality and equality constraints:

$$f(x) = \arg \min_{y \in \mathbb{R}^{n_{out}}} \quad \frac{1}{2} y^\top Q y + p^\top \sin(y)$$
$$\text{s.t.} \quad Gy \leq h, Cy = x,$$

where $Q \in \mathbb{R}^{n_{out} \times n_{out}}$, $p \in \mathbb{R}^{n_{out}}$, $G \in \mathbb{R}^{n_{ineq} \times n_{out}}$, $h \in \mathbb{R}^{n_{ineq}}$, $C \in \mathbb{R}^{n_{eq} \times n_{out}}$, and $x \in \mathbb{R}^{n_{eq}}$. Input $x$ is sampled in $[-1, 1]^{n_{eq}}$, and $G$ and $C$ are randomly generated. To ensure that this optimization function is feasible for any input $x$, $h$ is calculated as $h = \sum_j |(GC^\dagger)_{(:,j)}|$ by utilizing $|x_j| \leq 1$ and $Gy = GC^\dagger x \leq \sum_j |(GC^\dagger)_{(:,j)}|$ (Donti et al., 2021). This experiment is trained using unsupervised learning, where the training loss is the objective value of the optimization problem. We set $n_{in} = n_{eq} = 3$, $n_{out} = n_{ineq} = 5$. The detailed expressions of the constraints are provided in Appendix D.2. For this experiment, the benchmark optimizer is IPOPT, and we randomly generate 1000 training samples and 1000 testing samples. The training is performed for 10000 epochs with a batch size of 1000.

The results are summarized in Table 3. It can be observed that the benchmark optimizer and both CAffNet variants strictly satisfy all constraints, whereas all other methods have constraint violations. Both HardNet and CAffNet-FF have a small optimality gap relative to IPOPT, with CAffNet-FF obtaining the smallest gap on average. Further, note that equality constraints are more challenging to satisfy, such that NN and HardNet almost violate all equality constraints. Finally, when comparing the testing times, both CAffNet variants are observed to be significantly faster than the benchmark optimizer IPOPT while ensuring constraint satisfaction. We also implement a lite version of CAffNet-FF, which considers a reduced set of constraint combinations as described in Remark 3.7, and it achieves the same performance as the full CAffNet-FF while further reducing training and testing time by approximately 45% compared to the full constraint combination version.

## 4.3. Learning Control Policies in Safety-Critical Systems

In this experiment, we learn control policies for a unicycle model with input-dependent affine constraints for enforcing safety. Consider a general control-affine dynamical system of the form $\dot{x} = f(x) + g(x)u(x)$, where $x \in \mathbb{R}^{n_{in}}$ and

*Table 3.* Comparison of optimization solver learning methods with inequality and equality constraints (mean; standard deviation in parentheses). Both CAffNets strictly satisfy all constraints. CAffNet-FF has comparable objective values to the benchmark optimizer IPOPT. In addition to constraint satisfaction, both CAffNet variants also have significantly faster testing time than IPOPT, making them suitable for real-time applications. Both NN and HardNet have constraint violations, especially of the equality constraints.

| METHOD | OBJ. VALUE | INEQUALITY VIOLATION | | | EQ VIOL. | | | $T_{train}$ (MS) | $T_{test}$ (MS) |
|---|---|---|---|---|---|---|---|---|---|
| | | MAX | MEAN | NUM (%) | MAX | MEAN | NUM (%) | | |
| IPOPT | **-0.3494** | **0.0000** | **0.0000** | **0.00%** | **0.0000** | **0.0000** | **0.00%** | - | 19799.8000 |
| | (0.4995) | (0.0000) | (0.0000) | (0.00%) | (0.0000) | (0.0000) | (0.00%) | (-) | (295.16) |
| NN | -0.1194 | 0.0018 | 0.0004 | 0.31% | 0.1136 | 0.0685 | 100.00% | **4.2700** | **1.2300** |
| | (0.5824) | (0.0015) | (0.0003) | (0.21%) | (0.0035) | (0.0023) | (0.00%) | (0.09) | (0.01) |
| HARDNET | -0.3000 | 0.0027 | 0.0006 | 3.24% | 0.0435 | 0.0258 | 99.99% | 7.3800 | 2.1100 |
| | (0.4975) | (0.0026) | (0.0006) | (2.91%) | (0.0123) | (0.0071) | (0.01%) | (0.20) | (0.03) |
| CAFFNET-FF | -0.3418 | **0.0000** | **0.0000** | **0.00%** | **0.0000** | **0.0000** | **0.00%** | 1325.3800 | 1298.6700 |
| | (0.5028) | (0.0000) | (0.0000) | (0.00%) | (0.0000) | (0.0000) | (0.00%) | (26.28) | (26.28) |
| CAFFNET-FF (LITE) | -0.3418 | **0.0000** | **0.0000** | **0.00%** | **0.0000** | **0.0000** | **0.00%** | 737.1600 | 713.6900 |
| | (0.5028) | (0.0000) | (0.0000) | (0.00%) | (0.0000) | (0.0000) | (0.00%) | (20.54) | (20.73) |
| CAFFNET-TF | -0.1576 | **0.0000** | **0.0000** | **0.00%** | **0.0000** | **0.0000** | **0.00%** | 1327.7900 | 1299.9500 |
| | (0.3653) | (0.0000) | (0.0000) | (0.00%) | (0.0000) | (0.0000) | (0.00%) | (24.21) | (24.62) |

*Table 4.* Comparison of safe learning control policies for a single integrator system (mean; standard deviation in parentheses). CAffNet satisfies all constraints, while NN and HardNet incur constraint violations.

| METHOD | COST | INEQUALITY VIOLATION | | | $T_{train}$ (MS) | $T_{test}$ (MS) |
|---|---|---|---|---|---|---|
| | | MAX | MEAN | NUM (%) | | |
| NN | $4.1411 \times 10^5$ | 0.1348 | 0.0106 | 2.60% | **2895.1300** | **47.1700** |
| | $(6.7984 \times 10^4)$ | (0.0083) | (0.0006) | (0.03%) | (263.98) | (4.17) |
| HARDNET | $4.5701 \times 10^5$ | 0.1317 | 0.0104 | 2.60% | 4885.6000 | 855.5100 |
| | $(4.2737 \times 10^4)$ | (0.0066) | (0.0005) | (0.03%) | (231.74) | (81.92) |
| CAFFNET-FF | $7.2060 \times 10^5$ | **0.0000** | **0.0000** | **0.00%** | 12207.9300 | 1885.1000 |
| | $(6.9090 \times 10^4)$ | (0.0000) | (0.0000) | (0.00%) | (306.59) | (45.96) |

$u(x) \in \mathbb{R}^{n_{out}}$. The unicycle robot dynamics is as follows:

$$\dot{x} = \begin{bmatrix} 0 \\ 0 \\ 0 \end{bmatrix} + \begin{bmatrix} \cos(\theta) & 0 \\ \sin(\theta) & 0 \\ 0 & 1 \end{bmatrix} u(x),$$

with $x = [p_x, p_y, \theta]^\top$ and $u(x) = [v, \omega]^\top$. $(p_x, p_y)$ represent the position of the unicycle, $\theta$ is its orientation, $v$ is the linear velocity command, and $\omega$ is the angular velocity command. The nominal controller $u_{nom}(x)$ is a PID controller without considering safety conditions, and the neural network output is performed as a correction to the nominal control command, i.e., the overall command to the unicycle is chosen as $u(x) = u_{nom}(x) + u_{net}(x)$. The state constraints are $-5\ m \le p_x \le 1\ m$, $-4\ m \le p_y \le 2\ m$, and $-\pi\ rad \le \theta \le \pi\ rad$, while the control command constraints are $-0.01\ m/s \le v \le 1\ m/s$ and $-0.5\ rad/s \le \omega \le 0.5\ rad/s$. We saturate the nominal and unicycle control command to satisfy the control bounds, modeling realistic actuator saturation. The goal position is located at $(0, 0)$, and the agent is considered to have arrived once it enters a circle of radius $0.1\ m$. We set $\Delta t$ as $0.1\ s$, and randomly generated 300 initial states within the state constraints for training, with orientations towards the origin. For each epoch during the training, we simulate

for $15\ s$ and compute the cost that encourages the robot to move toward $(0, 0)$ while keeping both the state magnitude and the corrective control effort small:

$$J = \sum_{k=0}^{N-1} \left( x_k^\top Q x_k + u_{net}(x_k)^\top R u_{net}(x_k) \right) + x_N^\top Q_N x_N,$$

where $N$ is the total number of receding horizon steps, $Q = \text{diag}(1000, 1000, 0)$, $R = \text{diag}(1, 1)$, and $Q_N = \text{diag}(10^6, 10^6, 0)$. Moreover, we consider three convex obstacles with the hyperplane representation, i.e.,

$$\mathcal{O}_j = \{x \in \mathbb{R}^2 \mid A_j x \le b_j\}, \quad j \in J = \{1, 2, 3\}.$$

For each obstacle $\mathcal{O}_j$, each row of $A_j$ and $b_j$ represents a line that aligns with its corresponding edge, e.g., $a_j^i x = b_j^i$, where $i \in M_j = \{1, 2, \dots, m_j\}$ indicates the $i$-th edge, and $m_j$ is the total number of edges/constraints for obstacle $\mathcal{O}_j$. Thus, the halfspace being outside of the corresponding obstacle edge can be represented by a single control barrier function candidate $h_j^i(x) = a_j^i x - b_j^i \ge 0$. Then the collision-free condition is that the position lies outside at least one edge-aligned constraints, i.e., $\max_{i \in M_j} h_j^i(x) \ge 0$, which can be enforced by the union of $h_j^i(x)$ for each $i$-th edge of the obstacle $\mathcal{O}_j$. Thus, we use a smooth function over-approximation in (Molnar & Ames, 2023, Eq. (26)) to

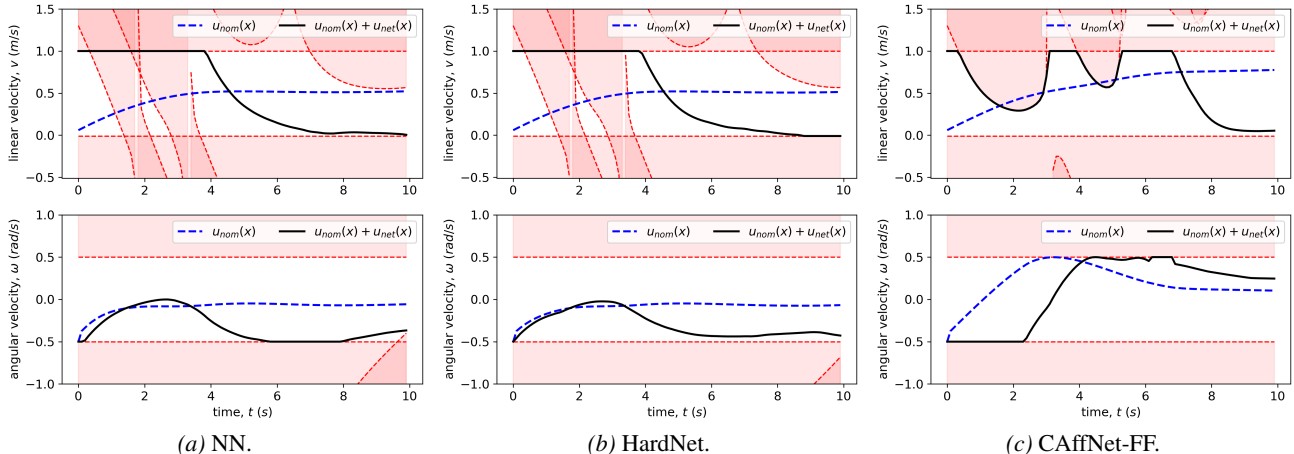

*Figure 7.* Comparison of control inputs for learning control policies in safety-critical systems. NN and HardNet choose a relatively high linear velocity in the beginning, even when close to an obstacle, leading to constraint violations in the linear velocity and consequently colliding with the obstacle. In contrast, CAffNet-FF selects safe control commands that satisfy all constraints throughout the trajectory.

convert $x \in \mathcal{X} \setminus \mathcal{O}_j$ into a single constraint $h_j(x)$:

$$h_j(x) = \frac{1}{\kappa} \ln\left(\sum_{i \in M_j} e^{\kappa h_j^i(x)}\right) - \frac{\eta_j}{\kappa},$$

where $\kappa = 10$ is a positive constant to adjust the tightness and smoothness of the over-approximation, and $\eta_j = \ln(m_j)$ is the over-approximation buffer. The smooth union approximation satisfies $h_j(x) \leq \max_{i \in M_j} h_j^i(x)$. Hence, the sufficient condition for collision avoidance is $h_j(x) \geq 0, \forall j \in J$. Safety (i.e., collision avoidance) is then enforced using control barrier functions (CBFs) (Ames et al., 2016) that take the following form $\forall j \in \{1, 2, 3\}$:

$$L_f h_j(x) + L_g h_j(x)u(x) \geq -\alpha(h_j(x)),$$

where $\alpha(\cdot)$ is a class $\mathcal{K}$ function that is strictly increasing with $\alpha(0) = 0$, and we set $\alpha(h(x)) = h(x)$ in this experiment. The Lie derivatives are given by

$$L_f h_j(x) = \sum_{i \in M_j} \lambda_i(x) L_f h_j^i(x),$$

$$L_g h_j(x) = \sum_{i \in M_j} \lambda_i(x) L_g h_j^i(x),$$

with $\lambda_j^i(x) = e^{\kappa(h_j^i(x) - h_j(x))}$. The state constraints $h_x(x) = b_x - A_x x \geq 0$ can also be similarly converted to input-dependent affine constraints with CBFs, and the control constraints are also assumed to be affine: $A_u u(x) \leq b_u$. Thus, the aggregated input-dependent affine constraint is in the form of (1) with

$$A(x) = \begin{bmatrix} -L_g h_1(x) \\ -L_g h_2(x) \\ -L_g h_3(x) \\ -L_g h_x(x) \\ A_u \end{bmatrix}, b(x) = \begin{bmatrix} L_f h_1(x) + \alpha(h_1(x)) \\ L_f h_2(x) + \alpha(h_2(x)) \\ L_f h_3(x) + \alpha(h_3(x)) \\ L_f h_x(x) + \alpha(h_x(x)) \\ b_u \end{bmatrix}.$$

In this experiment, for brevity, we omitted CAffNet-TF. Further details of the problem are given in Appendix D.3. From Figs. 7–6 and Table 4, we can observe that the unicycle robot using CAffNet-FF successfully arrives near the goal region without any collisions and control constraint violations, while NN and HardNet collide with obstacles $\mathcal{O}_1$ and $\mathcal{O}_3$.

Finally, to demonstrate the value of jointly training NNs with the CAffine projection layer, we also enforced the projection on the output of the unconstrained NN *a posteriori*, and found that in this case, the robot gets stuck near an obstacle and never makes it to the goal. Thus, CAffNet is trainable and can find the optimal feasible solution.

## 5. Conclusion

We introduced CAffNet, a novel neural network framework that guarantees hard satisfaction of input-dependent affine constraints with a closed-form architecture. By leveraging constraint decomposition and a trainable null-space projection, our method resolves limitations on constraint cardinality and ensures feasibility whenever the constraint/solution set is non-empty. Further, we proved that CAffNet is a universal approximator, allowing the underlying neural networks to retain their expressive power while adhering to (safety) constraints. Future work will focus on extending this framework to support non-affine constraints and validating its performance on physical robotic platforms. Improvement of the scalability of CAffNet is also an important future direction.

## Acknowledgements

This work was supported by the National Science Foundation (NSF) under Grant No. CNS-2313814, by the InnoCORE program of the Ministry of Science and ICT (#N10250155), and by the Institute of Information & Communications Technology Planning & Evaluation (IITP) grant funded by the Korean government (MSIT) (No.RS-2020-II201336, Artificial Intelligence Graduate School Program (UNIST)).

## Impact Statement

This work aims to improve the reliability of neural networks by embedding hard affine constraint satisfaction directly into the model architecture. Its potential positive impact is to support safer use of learned models in domains where outputs must satisfy operational or safety constraints, such as robotics, control, and other safety-critical systems. By providing formal guarantees under the stated assumptions, the proposed framework may reduce constraint violations compared with penalty-based training or post-processing methods.

At the same time, these guarantees apply only to the specified constraints and their feasibility assumptions. They do not by themselves ensure that the constraints are complete, that the system model is accurate, or that deployment conditions match the training and evaluation settings. Misuse or overreliance on the method could therefore lead to unsafe behavior if relevant hazards are omitted or incorrectly modeled. Deployment in safety-critical applications should include independent validation, monitoring, and domain-specific safety analysis.

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

## A. Proof of Lemma 3.3 (Existence of a Feasible Solution)

*Proof.* By Assumption 3.2, the feasible set $\mathcal{S}(x)$ is a non-empty (convex) polyhedron. Then, there exists at least one minimal face $F(x)$ of $\mathcal{S}(x)$ (cf. Definition 3.1), such that $F(x) = \{y \in \mathbb{R}^n \mid A_{\gamma^*}(x)y = b_{\gamma^*}(x)\}$, where $A_{\gamma^*}(x)$ and $b_{\gamma^*}(x)$ is a sub-constraint of $A(x)$ and $b(x)$ corresponding to the active constraints that define the minimal face $F(x)$. Furthermore, since $A_{\gamma^*}(x)$ and $b_{\gamma^*}(x)$ form a minimal face, it is formed by $k^*$ linearly independent constraints and $k^* \leq n_{out}$. In addition, $k^* \leq m$ as there are at most $m$ constraints to choose from. Thus, since CAffNet constructs the set $\Gamma$ by choosing all combinations of indices of size up to $\min(m, n_{out})$, this specific index set $\gamma^*$ with corresponding $k^*$ constraints is guaranteed to be within $\Gamma$. Then, for this index set $\gamma^*$, CAffNet generates a candidate $\mathcal{P}_{\gamma^*}(x)$ via the projection:

$$\mathcal{P}_{\gamma^*}(x) = f_\theta(x) - A_{\gamma^*}^\dagger(x)(A_{\gamma^*}(x)f_\theta(x) - b_{\gamma^*}(x)) + (I - A_{\gamma^*}^\dagger(x)A_{\gamma^*}(x))w_\phi(x). \tag{7}$$

We now verify that this candidate satisfies the equality constraints of the minimal face. For brevity, we omit the argument $x$ in the rest of this derivation. Multiplying (7) by $A_{\gamma^*}$ and using the property of Moore-Penrose pseudoinverses that $AA^\dagger A = A$ holds, we have

$$A_{\gamma^*}\mathcal{P}_{\gamma^*} = A_{\gamma^*}f_\theta - A_{\gamma^*}A_{\gamma^*}^\dagger(A_{\gamma^*}f_\theta - b_{\gamma^*}) + A_{\gamma^*}(I - A_{\gamma^*}^\dagger A_{\gamma^*})w_\phi \tag{8}$$

$$= A_{\gamma^*}A_{\gamma^*}^\dagger b_{\gamma^*}. \tag{9}$$

Since $F(x)$ is a non-empty face, the system $A_{\gamma^*}y = b_{\gamma^*}$ has at least one solution. A necessary and sufficient condition for the consistency of a linear system $Ax = b$ is $AA^\dagger b = b$. Therefore, we obtain

$$A_{\gamma^*}\mathcal{P}_{\gamma^*} = b_{\gamma^*}. \tag{10}$$

This implies that $\mathcal{P}_{\gamma^*}(x) \in F(x)$. Since $F(x)$ is a face of $\mathcal{S}(x)$, it is a subset of $\mathcal{S}(x)$ ($F(x) \subseteq \mathcal{S}(x)$). Consequently, any point in $F(x)$ satisfies the full set of inequalities, i.e.,

$$A(x)\mathcal{P}_{\gamma^*}(x) \leq b(x). \tag{11}$$

Thus, the projection set $\{\mathcal{P}_\gamma(x) \mid \gamma \in \Gamma\}$ contains at least one feasible solution $\mathcal{P}_{\gamma^*}(x)$ satisfying the constraints. $\square$

## B. Proof of Theorem 3.4 (Hard Constraint Satisfaction of CAffNet)

*Proof.* For any input $x \in \mathbb{R}^{n_{in}}$, under Assumption 3.2, the feasible region $\mathcal{S}(x) = \{y \in \mathbb{R}^{n_{out}} \mid A(x)y \leq b(x)\}$ is non-empty. Further, by Lemma 3.3, there exists at least one candidate $y \in \{\mathcal{P}_\gamma(x) \mid \gamma \in \Gamma\}$, such that $A(x)y \leq b(x)$. By the definition of the candidate set $\mathcal{S}_\mathcal{P}(x)$ in (5), this implies $y \in \mathcal{S}_\mathcal{P}(x)$, and therefore $\mathcal{S}_\mathcal{P}(x)$ is non-empty. And from (6), the CAffNet output $\mathcal{P}^*(x)$ is either $f_\theta(x)$ if it satisfies the input-dependent affine constraints, or one of the candidates in $\mathcal{S}_\mathcal{P}(x)$. In both cases, the output $\mathcal{P}^*(x)$ satisfies the constraints. $\square$

## C. Proof of Theorem 3.5 (Universal Approximation of CAffNet)

*Proof.* Given the safe region $\mathcal{S}(x) := \{y \in \mathbb{R}^{n_{out}} \mid A(x)y \leq b(x)\}$, let $f_t \in \mathcal{F}_{target}$ be any target function satisfying the constraints, i.e., $f_t(x) \in \mathcal{S}(x)$. For any $\epsilon > 0$, since (an unconstrained) $\mathcal{F}_\theta$ universally approximates $\mathcal{F}$ for any norm $p \in [1, \infty)$, there exists a network $f_\theta \in \mathcal{F}_\theta$ such that $\|f_\theta(x) - f_t(x)\|_p < K$, where $K = \dfrac{\epsilon}{3 + 3\sqrt{n_{out}}}$ is a constant. We now prove that the CAffNet output $\mathcal{P}^*(x)$ satisfies $\|\mathcal{P}^*(x) - f_t(x)\|_p < \epsilon$, i.e., inherits the universal approximation property.

We analyze the CAffNet output $\mathcal{P}^*(x)$ for two cases:

**Case 1:** $f_\theta(x) \in \mathcal{S}(x)$.
By the definition of the CAffNet projection in (6), if the argument is feasible, CAffNet outputs the argument itself:

$$\mathcal{P}^*(x) = f_\theta(x). \tag{12}$$

Thus, the approximation error is strictly bounded by the (unconstrained) approximation error, i.e.,

$$\|\mathcal{P}^*(x) - f_t(x)\|_p = \|f_\theta(x) - f_t(x)\|_p < K < \epsilon. \tag{13}$$

**Case 2:** $f_\theta(x) \notin \mathcal{S}(x)$.

Since $f_t(x) \in \mathcal{S}(x)$ and $f_\theta(x) \notin \mathcal{S}(x)$, the line that connects $f_\theta(x)$ and $f_t(x)$ must intersect with the boundary of $\mathcal{S}(x)$ at least once. Suppose that the intersection point is $f_\gamma(x)$, which satisfies a sub-constraint defined by the index set $\gamma$, i.e., $A_\gamma f_\gamma(x) = b_\gamma(x)$. As $f_\gamma(x)$ is between $f_\theta(x)$ and $f_t(x)$, we have $f_\gamma(x) = \delta f_t(x) + (1 - \delta) f_\theta(x)$ for some $\delta \in (0, 1]$. Then

$$
\begin{aligned}
\|f_\gamma(x) - f_\theta(x)\|_p &= \|\delta f_t(x) + (1 - \delta) f_\theta(x) - f_\theta(x)\|_p \\
&= \|\delta(f_t(x) - f_\theta(x))\|_p \\
&\leq \|f_t(x) - f_\theta(x)\|_p < K.
\end{aligned}
\tag{14}
$$

Let the target combination-specific null-space vector be $w_{t,\gamma}(x)$ such that $\|w_{t,\gamma}(x)\|_p \leq \|f_\theta(x) - f_t(x)\|_p < K$, which always exists since $w_{t,\gamma}(x) = 0$ satisfies the inequality. Additionally, there also exists a single continuous target function $w_t(x)$ such that $\|w_t(x)\|_p \leq \|w_{t,\gamma}(x)\|_p$ for all combinations $\gamma$, which again exists since $w_t(x) = 0$ satisfies the inequality. From the universal approximation theorem of NNs for the null space term in (4), there exists a learned null-space vector $w_\phi(x)$ that satisfies $\|w_\phi(x) - w_t(x)\|_p < K$, from which we can bound the learned null-space vector as

$$
\begin{aligned}
\|w_\phi(x)\|_p &= \|w_\phi(x) - w_t(x) + w_t(x)\|_p \\
&\leq \|w_\phi(x) - w_t(x)\|_p + \|w_t(x)\|_p \\
&\leq \|w_\phi(x) - w_t(x)\|_p + \|w_{t,\gamma}(x)\|_p \\
&< 2K.
\end{aligned}
\tag{15}
$$

We then first evaluate bounds of $A_\gamma^\dagger A_\gamma$ and $I - A_\gamma^\dagger A_\gamma$. Given the definition of the matrix norm $\|A\|_p = \max_{v \neq 0} \frac{\|Av\|_p}{\|v\|_p}$ for a matrix $A \in \mathbb{R}^{m \times n}$, we utilize the norm equivalence of a vector $v \in \mathbb{R}^n$ that $\|v\|_q \leq \|v\|_r \leq n^{\frac{1}{r} - \frac{1}{q}} \|v\|_q$, for norm $1 \leq r \leq q$. If $p = 1$, let $r = 1$, $q = 2$ and then we have

$$
\|v\|_2 \leq \|v\|_1 \leq \sqrt{n} \|v\|_2.
$$

Applying this vector norm to the matrix norm, we have

$$
\|A\|_1 = \max_{v \neq 0} \frac{\|Av\|_1}{\|v\|_1} \leq \max_{v \neq 0} \frac{\sqrt{m}\|Av\|_2}{\|v\|_2} = \sqrt{m} \max_{v \neq 0} \frac{\|Av\|_2}{\|v\|_2} = \sqrt{m}\|A\|_2.
$$

If $p \geq 2$, let $r = 2$, $q = p$ and then we have

$$
\|v\|_p \leq \|v\|_2 \leq n^{\frac{1}{2} - \frac{1}{p}} \|v\|_p \leq \sqrt{n}\|v\|_p.
$$

Applying this vector norm to the matrix norm, we have

$$
\|A\|_p = \max_{v \neq 0} \frac{\|Av\|_p}{\|v\|_p} \leq \max_{v \neq 0} \frac{\sqrt{n}\|Av\|_2}{\|v\|_2} = \sqrt{n} \max_{v \neq 0} \frac{\|Av\|_2}{\|v\|_2} = \sqrt{n}\|A\|_2.
$$

Since $A_\gamma^\dagger A_\gamma$ and $I - A_\gamma^\dagger A_\gamma$ are square matrices in $\mathbb{R}^{n_{out} \times n_{out}}$, as well as from the properties of Moore-Penrose pseudoinverses that $\|A_\gamma^\dagger A_\gamma\|_2 \leq 1$ and $\|I - A_\gamma^\dagger A_\gamma\|_2 \leq 1$, we have

$$
\begin{aligned}
\|A_\gamma^\dagger A_\gamma\|_p &\leq \sqrt{n_{out}}\|A_\gamma^\dagger A_\gamma\|_2 \leq \sqrt{n_{out}}, \\
\|I - A_\gamma^\dagger A_\gamma\|_p &\leq \sqrt{n_{out}}\|\mathcal{I} - A_\gamma^\dagger A_\gamma\|_2 \leq \sqrt{n_{out}}.
\end{aligned}
\tag{16}
$$

Now we analyze the distance between the corresponding projection $\mathcal{P}_\gamma(x)$ in (4) and the target, which can be bounded as follows:

$$
\begin{aligned}
\|f_t - \mathcal{P}_\gamma\|_p &= \|f_t - \left(f_\theta - A_\gamma^\dagger(A_\gamma f_\theta - b_\gamma) + (I - A_\gamma^\dagger A_\gamma)w_\phi\right)\|_p \\
&= \|f_t - \left(f_\theta - A_\gamma^\dagger(A_\gamma f_\theta - A_\gamma f_\gamma) + (I - A_\gamma^\dagger A_\gamma)w_\phi\right)\|_p \\
&= \|(f_t - f_\theta) + A_\gamma^\dagger A_\gamma(f_\theta - f_\gamma) - (I - A_\gamma^\dagger A_\gamma)w_\phi\|_p \\
&\leq \|f_t - f_\theta\|_p + \|A_\gamma^\dagger A_\gamma\|_p\|f_\theta - f_\gamma\|_p + \|I - A_\gamma^\dagger A_\gamma\|_p\|w_\phi\|_p \\
&\leq \|f_t - f_\theta\|_p + \sqrt{n_{out}}\|f_\theta - f_t\|_p + \sqrt{n_{out}}\|w_\phi\|_p \\
&< (1 + 3\sqrt{n_{out}})K.
\end{aligned}
$$

where, for brevity, we simplify the input-dependent notation by omitting the argument $x$. The second equality is obtained since the intersection point $f_\gamma$ satisfies $b_\gamma = A_\gamma f_\gamma$, while the first inequality is obtained using triangle inequality. The second inequality follows from the inequality in (14) as well as the matrix norm bounds in (16). Finally, the overall bound is obtained based on the distance between $f_t$ and $f_\theta$ by the universal approximation property and the learned null-space function bound in (15).

Further, since $\mathcal{P}^*(x)$ in (6) is selected as the closest feasible candidate to $f_\theta(x)$, we have

$$
\begin{aligned}
\|\mathcal{P}^* - f_\theta\|_p &\leq \|\mathcal{P}_\gamma - f_\theta\|_p \\
&= \|\mathcal{P}_\gamma - f_t + f_t - f_\theta\|_p \\
&\leq \|\mathcal{P}_\gamma - f_t\|_p + \|f_t - f_\theta\|_p \\
&< (2 + 3\sqrt{n_{out}})K,
\end{aligned}
$$

which implies that

$$
\begin{aligned}
\|\mathcal{P}^* - f_t\|_p &= \|\mathcal{P}^* - f_\theta + f_\theta - f_t\|_p \\
&\leq \|\mathcal{P}^* - f_\theta\|_p + \|f_\theta - f_t\|_p \\
&< (3 + 3\sqrt{n_{out}})K = \epsilon.
\end{aligned}
$$

Thus, CAffNet preserves the universal approximation property. $\qquad\square$

## D. Experimental Details

### D.1. Details for Learning Nonlinear Function with Piecewise Constraints

The piecewise target function is defined as:

$$
f(x) = \begin{cases}
-5\sin(\frac{\pi}{2}(x+1)) - 2, & \text{if } x \in (-\infty, -1], \\
-2, & \text{if } x \in (-1, 0], \\
2 - 9(x - \frac{2}{3})^2, & \text{if } x \in (0, 1], \\
\frac{3}{x^2} - 2. & \text{if } x \in (1, \infty).
\end{cases}
$$

There are two piecewise upper bounds:

$$
g_1^u(x) = \begin{cases}
-3\sin(\frac{\pi}{2}(x+1)) + \frac{1}{5}, & \text{if } x \in (-\infty, -1], \\
-2, & \text{if } x \in (-1, 0], \\
3 - 4(x - \frac{1}{2})^2, & \text{if } x \in (0, 1], \\
2, & \text{if } x \in (1, \infty),
\end{cases}
\quad,\quad
g_2^u(x) = \begin{cases}
-3\sin(\frac{\pi}{2}(x+1))^3 + 1, & \text{if } x \in (-\infty, -1], \\
2, & \text{if } x \in (-1, 0], \\
3 - 4(x - \frac{4}{5})^2, & \text{if } x \in (0, 1], \\
2.5, & \text{if } x \in (1, \infty),
\end{cases}
$$

as well as two piecewise lower bounds:

$$
g_1^l(x) = \begin{cases}
5\sin(\frac{\pi}{2}(x+1))^2 - 3, & x \in (-\infty, -1], \\
-2, & x \in (-1, 0], \\
(4 - 9(x - \frac{2}{3})^2)x - \frac{5}{2}, & x \in (0, 1], \\
\frac{3}{x^3} - \frac{5}{2}, & x \in (1, \infty),
\end{cases}
\quad,\quad
g_2^l(x) = \begin{cases}
5\sin(\frac{\pi}{2}(x+1))^8 - 2, & \text{if } x \in (-\infty, -1], \\
-3, & \text{if } x \in (-1, 0], \\
(5 - 4(x - \frac{1}{6})^2)x - \frac{5}{2}, & \text{if } x \in (0, 1], \\
\frac{3}{2x^3} - \frac{16}{9}, & \text{if } x \in (1, \infty).
\end{cases}
$$

From the above, the (two-sided) affine constraints for HardNet are

$$
b^l(x) = \begin{bmatrix} g_1^l(x) \\ g_2^l(x) \end{bmatrix}, \quad A(x) = \begin{bmatrix} 1 \\ 1 \end{bmatrix}, \quad b^u(x) = \begin{bmatrix} g_1^u(x) \\ g_2^u(x) \end{bmatrix},
$$

and equivalently, the affine constraints for all other methods are

$$
A(x) = \begin{bmatrix} 1 \\ 1 \\ -1 \\ -1 \end{bmatrix}, \quad b(x) = \begin{bmatrix} g_1^u(x) \\ g_2^u(x) \\ -g_1^l(x) \\ -g_2^l(x) \end{bmatrix}.
$$

### D.2. Details for Learning Optimization Solver

For HardNet, we need to concatenate the constraints as upper and lower bounds:

$$b^l(x) = \begin{bmatrix} -M\mathbb{1}_{n_{ineq}} \\ x \end{bmatrix}, \quad A(x) = \begin{bmatrix} G \\ C \end{bmatrix}, \quad b^u(x) = \begin{bmatrix} h \\ x \end{bmatrix},$$

where $M$ is a large positive constant, while for the other approaches, the constraints are

$$A(x) = \begin{bmatrix} G \\ C \\ -C \end{bmatrix}, \quad b(x) = \begin{bmatrix} h \\ x \\ -x \end{bmatrix}.$$

### D.3. Details for Learning Control Policies for Safety-critical Systems

With the variable ranges given in Section 4.3, the state constraints are

$$A_x = \begin{bmatrix} 1 & 0 & 0 \\ -1 & 0 & 0 \\ 0 & 1 & 0 \\ 0 & -1 & 0 \\ 0 & 0 & 1 \\ 0 & 0 & -1 \end{bmatrix}, \quad b_x = \begin{bmatrix} 1 \\ 5 \\ 2 \\ 4 \\ \pi \\ \pi \end{bmatrix},$$

and the control input constraints are

$$A_u = \begin{bmatrix} 1 & 0 \\ -1 & 0 \\ 0 & 1 \\ 0 & -1 \end{bmatrix}, \quad b_u = \begin{bmatrix} 1 \\ 0.01 \\ 0.5 \\ 0.5 \end{bmatrix}.$$

Moreover, the three obstacles placed in the environment are $\mathcal{O}_j = \{x \in \mathbb{R}^2 \mid A_j x \le b_j\}, j \in J = \{1, 2, 3\}$, where $A_j$ and $b_j$ are given by

$$A_1 = \begin{bmatrix} 0.4472 & -0.8944 \\ 0.7071 & 0.7071 \\ -0.2425 & 0.9701 \\ -0.7071 & -0.7071 \\ -0.8944 & -0.4472 \end{bmatrix}, \quad b_1 = \begin{bmatrix} -0.2184 \\ -0.5303 \\ 0.6219 \\ 1.1667 \\ 1.4368 \end{bmatrix},$$

$$A_2 = \begin{bmatrix} -0.9685 & 0.2489 \\ 0.9417 & 0.3363 \\ -0.3714 & 0.9285 \\ 0.3714 & 0.9285 \\ -0.9417 & -0.3363 \\ -0.2976 & -0.9547 \end{bmatrix}, \quad b_2 = \begin{bmatrix} 1.2755 \\ -1.7670 \\ -0.8511 \\ -2.0249 \\ 2.3274 \\ 2.6868 \end{bmatrix},$$

$$A_3 = \begin{bmatrix} -0.9191 & 0.3939 \\ 0.8944 & 0.4472 \\ 0.9703 & -0.2419 \\ -0.8701 & -0.4930 \\ 0.0000 & -1.0000 \end{bmatrix}, \quad b_3 = \begin{bmatrix} 2.9916 \\ -1.9975 \\ -2.5305 \\ 2.4854 \\ 0.1000 \end{bmatrix}.$$

Fig. 8 illustrates the coordinate transformation used to compute the tracking error in the robot body frame. The displacement from the robot position $(p_x, p_y)$ to the reference position $(p_{x,ref}, p_{y,ref})$ is decomposed along the body-frame longitudinal and lateral axes, yielding the longitudinal error $e_{long}(t)$ and lateral error $e_{lat}(t)$ used by the nominal controller. The state error $e(t) = [e_{long}(t), e_{lat}(t), e_\theta(t)]^T$ is then given by

$$e(t) = \begin{bmatrix} \cos(\theta) & \sin(\theta) & 0 \\ -\sin(\theta) & \cos(\theta) & 0 \\ 0 & 0 & 1 \end{bmatrix} (x_{ref}(t) - x(t)).$$

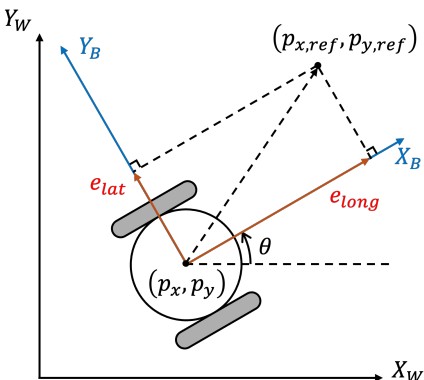

*Figure 8.* Illustration of the global-to-local coordinate transformation for the unicycle model. The displacement from the robot position $(p_x, p_y)$ to the reference position $(p_{x,ref}, p_{y,ref})$ in the world frame $X_W$-$Y_W$ is projected onto the body-frame longitudinal axis $X_B$ and lateral axis $Y_B$, yielding the tracking errors $e_{\text{long}}$ and $e_{\text{lat}}$, respectively.

Specifically, the nominal PID control input consists of the linear velocity command and the angular velocity command, where the former is computed based on the longitudinal distance error $e_{long}(t)$, and the latter is computed based on the lateral distance error $e_{lat}(t)$ and the heading angle error $e_\theta(t)$. The nominal input is given by

$$u_{nom} = \begin{bmatrix} 1 & 0 & 0 \\ 0 & 1 & 1 \end{bmatrix} \left( K_p e(t) + K_i \int_0^t e(\tau) d\tau + K_d \dot{e}(t) \right),$$

with parameters $K_p = \text{diag}(0.01, 0.2, 0)$, $K_i = \text{diag}(0.05, 0.005, 0)$, and $K_d = \text{diag}(0, 0.01, 0)$.

In addition, the initial state for testing is $x_0 = [-4.5, 0, 0.5]^\top$.

