# OpenReview forum: "CAffNet: Hard Constraint-Affine Neural Networks"
_ICML.cc/2026/Conference — ICML 2026 regular_

### Official Review · Reviewer_vrmx · 2026-03-06

**Soundness:** 3
**Presentation:** 4
**Significance:** 3
**Originality:** 3
**Overall Recommendation:** 4
**Confidence:** 4

**Summary:**

The paper proposes a differentiable projection module that is appended to a neural network to ensure (by construction) the satisfaction of input-dependent affine constraints on the output of the neural network.

**Compliance With Llm Reviewing Policy:**

Affirmed.

**Final Justification:**

The rebuttal adequately addresses my questions. However, the core scalability concern is only theoretically mitigated, but not empirically validated. I maintain my weak accept score.

**Key Questions For Authors:**

1. The projection requires computing the pseudo-inverse for all combinations of constraints (Eq. 4). Please discuss the scalability of the approach and provide a complexity analysis of the projection.

2. In the evaluation, the approach is used to learn safe control policies. How does the approach compare to safe Reinforcement Learning approaches (e.g., action masking [1])?

3. Are the two neural networks (unconstrained and null-space component) independent of each other? In Fig. 4, $f_\theta$ and $w_\theta$ have the same parameters $\theta$; however, in Sec. 3, "two neural networks" are mentioned.

4. The projections for different sub-constraints are independent, and each projection only uses matrix operations. Thus, the projection module should significantly benefit from GPU acceleration. Can corresponding results be included?

[1] Krasowski et al. "Provably Safe Reinforcement Learning: Conceptual Analysis, Survey, and Benchmarking". TMLR (2024)

**Limitations:**

No, the paper does not explicitly mention the limitations of their proposed approach.

**Strengths And Weaknesses:**

**Strengths:**

- The approach improves and generalizes on previous works (soft constraints and HardNet).
- The approach explicitly considers the case where the constraints admit nonunique solutions. In this case, the projection is trainable via the null-space component.
- The paper provides a formally rigorous derivation of the projection module and its analytical solution.
- The universal approximation theorem still holds for constrained neural networks.
- Overall, the paper is very well-written and well-structured.

**Weaknesses:**

- The paper does not mention the scalability of the approach.
- The evaluation only uses small neural networks (3x200 neurons) and transformers on toy examples.

---

> ### Author Rebuttal · Authors · 2026-03-31
>
> Thank you for the detailed review and insightful feedback. We have carefully considered your comments and address each concern below.
>
> >Scalability: Combinatorial explosion
>
> We apologize for omitting a discussion of our approach's limitations due to space constraints. As noted by other reviewers as well, the potential combinatorial explosion in the projection operator is indeed a recognized bottleneck, and we are working on an extension to address it. Please see our detailed response to Reviewer GcyK, where we also outline a potential solution.
>
> >Scalability: High-dimensional/large-scale problems
>
> Thank you for raising this point. We selected these toy examples and small networks primarily to align with established benchmark setups and ensure fair comparison. Evaluating on larger-scale, real-world systems is a planned direction for future work. Please see our detailed response to Reviewer GcyK.
>
> >Scalability: Complexity of pseudoinverse
>
> Thank you for your comment. The reviewer is right to observe that the computation of pseudoinverses can be expensive for large matrices. The maximum size of the matrices for which pseudoinverses need to be computed with our approach is $[min(m, n_{out}), n_{out}]$, which is often manageable since the network output tends to be much smaller than the network size (which does not affect the size of the matrix), and this computation is also highly optimized with many specialized algorithms for quickly finding pseudoinverses, such as PyTorch/CUDA via batched operations. In addition, for problems where the training data is available a priori, both $A_\gamma(x)$ and its pseudoinverse can be pre-computed for each data point and stored in a lookup table, which can be directly used during training and inference to reduce computation. We hope to include a discussion on this in the final revision, should the paper be accepted.
>
> >Scalability: Parallelization
>
> Thank you for suggesting the parallelization of the projection module. This is indeed feasible, as the matrix operations are independent and the module should in principle benefit from GPU acceleration. That said, for the smaller benchmark examples in our paper, the expected speedup is modest, and we have not yet explored this for larger problems. We would nevertheless like to include a discussion on GPU parallelization, and will aim to provide supporting results in the final version.
>
> >Comparison with safe reinforcement learning
>
> Thank you for asking about the safe control policy experiment and comparisons with safe reinforcement learning. The primary difference between our approach and safe RL is analogous to the difference between safe imitation learning and safe reinforcement learning: our approach assumes access to expert data or a nominal controller, rather than learning through trial-and-error exploration. Specifically, our experiment uses a nominal PID controller that is not necessarily safe, which is then made safe via our projection module based on control barrier functions. In contrast, provably safe RL methods, including the one you recommended, rely on neither expert data nor nominal controllers.
>
> We do intend to extend our approach to provably safe RL in future work. Of the three paradigms discussed in the recommended paper (Krasowski et al., 2024), namely action replacement, action projection, and action masking, action projection is the most natural fit, given that our approach is projection-based. Action replacement with a random safe action risks suboptimal performance, while action masking requires the prior elimination of unsafe actions, which can be computationally expensive in continuous control spaces, as the authors of that paper themselves acknowledge.
>
> >Two neural networks $f_\theta$ and $w_\theta$
>
> Thank you for pointing this out and we apologize for the oversight.
> You are correct that it should be two separate neural networks $f_{\theta}$ and $w_\phi$, with independent parameters $\theta$ and $\phi$, which we will correct in the final version.
>
> We hope that our responses have adequately addressed your concerns, and would warmly welcome any revision to the scores you deem appropriate. We sincerely appreciate the thorough and constructive feedback on our paper.

---

> > ### Author Rebuttal · Reviewer_vrmx · 2026-04-01
> >
> > All my questions have been sufficiently answered.

---

> > > ### Author Response · Authors · 2026-04-07
> > >
> > > Thank you for your review and for confirming that your concerns have been fully addressed.  We deeply appreciate your time, engagement, and thoughtful questions, which have helped strengthen our paper, and we would warmly welcome any revision to the scores you deem appropriate.

---

### Official Review · Reviewer_sLuW · 2026-03-09

**Soundness:** 1
**Presentation:** 2
**Significance:** 3
**Originality:** 3
**Overall Recommendation:** 4
**Confidence:** 4

**Summary:**

This paper introduced a neural network framework that guarantees hard satisfaction of input-dependent affine constraints. For an arbitrary number of constraints, the proposed CAffNet leverages constraint decomposition and a trainable null-space projection to ensure feasibility. The output of CAffNet picks up the original output of the unconstrained NN if it satisfies the constraints, or selects the projection with the minimum distance to the original output. The decomposition combinations are finite, and a null-space component is learned to further improve the projection's performance.

**Compliance With Llm Reviewing Policy:**

Affirmed.

**Final Justification:**

After reading the rebuttal and follow-up response, I believe my main concerns have been largely addressed. Although the presentation could still be improved and a major revision should be made for this paper, I now view the paper as technically sound and worthy of a higher score.

**Key Questions For Authors:**

Please see Weaknesses.

**Limitations:**

The theoretical claims appear overstated, and the proofs need substantial tightening.

**Strengths And Weaknesses:**

**Strengths**
1. The method is suitable for an arbitrary number of constraints.
2. The method can always ensure constraint satisfaction.

**Weaknesses**
1. The number of constraint subsets grows combinatorially with the number of constraints. It is unclear how to deal with a large $\Gamma$ and the resulting combinatorial explosion. Moreover, the learned null-space component may rely on a specific combination. It could be difficult to learn such a modifier when the final output is selected through the argmin operator in Equation (6).
2. Lemma 3.3 is not fully justified; the set F defined as $F=\\{y|A'y=b'\\}$ may correspond to an affine subspace that extends beyond the actual face of the polyhedron. Meanwhile, the proof in Appendix C uses $||A_\gamma^\dagger A_\gamma||_p\le1$, which does not seem to hold in general beyond the Euclidean norm.
3. The experiments are weak. How about the performance of the proposed method on a large-scale real-world optimization problem?

---

> ### Author Rebuttal · Authors · 2026-03-31
>
> Thank you for your time and effort in reviewing our paper and for the insightful feedback. We have carefully considered your comments and address each concern below.
>
> >Scalability: Combinatorial explosion
>
> Thank you for raising this point. We fully acknowledge the combinatorial explosion and scalability limitations that arise as the number of constraints increases. We are working on an extension to address this; please see our detailed response to Reviewer GcyK, where we also outline a potential solution.
>
> >Potential difficulties for learning null-space component
>
> Thank you for this insightful comment. You are correct that the single null-space component $w_\theta(x)$ must learn to generalize across different constraint combinations and this may be limiting. Theoretically, for highly complex feasible regions, the architecture can naturally be extended to utilize multiple null-space components, $w_{\gamma, \theta}(x)$, one for each subconstraint combination. However, the overall size of the network will accordingly be larger. This introduces a clear trade-off: utilizing $w_{\gamma, \theta}(x)$ could improve transient training performance and capacity, at the cost of increased network complexity and a higher parameter count. In our revised manuscript, we plan to formally introduce $w_{\gamma, \theta}(x)$ as an architectural variant for more complex domains and explicitly discuss this trade-off, should the paper be accepted. It is worth noting, however, that with either choice of the null-space component(s), it can be easily shown that the Universal Approximation property in Theorem 3.5 still holds.
>
> >Lemma 3.3 is not fully justified
>
> Thank you for raising this concern, and we apologize for this source of confusion that may have led to a misunderstanding. You are absolutely right that when projecting onto the set $F=\\{y \mid A'y=b'\\}$, there is a possibility of projecting onto an affine subspace that extends beyond the actual (minimal) face of the polyhedron. In fact, this was the primary reason that we added the additional $A(x)y\le b(x)$ constraint in Eq. (5) to rule out such a projection. Moreover, whether or not this projection is on an actual face or ruled out for being beyond the actual face, the feasibility result in Lemma 3.3 still holds because there will be always at least one other feasible projection. The reason for this is because our combinations are based on the definition of a minimal face in Definition 3.1 from (Schrijver, 1998,
> Theorem 8.4) that includes vertices of the polyhedron for which the projection is unique (with no null space) and directly onto the vertex itself, which by definition is a feasible candidate solution. We will revise our paper to make this point clearer, should our paper be accepted.
>
> >Proof in Appendix C
>
> We thank the reviewer for their careful reading and apologize for the oversight. The reviewer is correct that $\lVert A_\gamma^\dagger A_\gamma \rVert_p \leq 1$ only holds when $p=2$. Nonetheless, the Universal Approximation property still holds for any $p$ with some minor modifications of the proof by leveraging norm equivalence.
>
> Specifically, by norm equivalence and $\lVert A_\gamma^\dagger A_\gamma\rVert_2 \le 1$,  it can be shown that $\lVert A_\gamma^\dagger A_\gamma\rVert_p \le \sqrt{n_{out}}$ holds for any $p$. Hence, if we revise the threshold of $\frac{\epsilon}{6}$ in Eq. (13) to $\frac{\epsilon}{3+3\sqrt{n_{out}}}$, such that $\lVert f_\theta(x) - f_t(x)\rVert_p < \frac{\epsilon}{3+3\sqrt{n_{out}}}
> $ and $\lVert \omega_\theta(x) - \omega_t(x)\rVert_p < \frac{\epsilon}{3+3\sqrt{n_{out}}}$. Then, Eq. (14) becomes $\lVert f_\gamma(x) - f_\theta(x)\rVert_p < \frac{\epsilon}{3+3\sqrt{n_{out}}}$ and further,  $\lVert f_t - \mathcal{P}^{(\gamma)}\rVert_p < \frac{(1+3\sqrt{n_{out}})\epsilon}{3+3\sqrt{n_{out}}}$ and  $\lVert \mathcal{P}^* - f_\theta\rVert_p < \frac{(2+3\sqrt{n_{out}})\epsilon}{3+3\sqrt{n_{out}}}$. As a result,  $\lVert\mathcal{P}^* - f_t\rVert_p < \epsilon$ holds, which implies the universal approximation property still holds. We will revise the proof in Appendix C accordingly (and provide more details) in the revised manuscript.
>
> >Scalability: High-dimensional/large-scale problems
>
> Thank you for raising this point. We selected these smaller-scale experiments primarily to align with benchmark paper setups and ensure fair comparison. Evaluating on larger-scale, real-world systems is a planned direction for future work. Please see our detailed response to Reviewer GcyK.
>
> We hope that our responses have adequately addressed your concerns, and would warmly welcome any revision to the scores you deem appropriate. We sincerely appreciate the thorough and constructive feedback on our paper.

---

> > ### Author Rebuttal · Reviewer_sLuW · 2026-04-02
> >
> > I appreciate the clarification, but I still find the treatment of the null-space component under-explained.
> >
> > The current model shares a single $\(w_\theta(x)\)$ across all sub-constraint combinations, while a combination-specific $\(w_{\gamma,\theta}(x)\)$ would appear too costly. What is the actual mechanism that makes the shared $\(w_\theta(x)\)$ learnable and stable across changing combinations in this paper and in the current experiment? Any tricks or insights would be appreciated.

---

> > > ### Author Response · Authors · 2026-04-07
> > >
> > > We are glad to hear that most of your concerns have been addressed, and thank you for your insightful follow-up question. We apologize that the treatment of the null-space component is still under-explained (due to the limited characters for rebuttals), and we hope to do a better job at providing the missing details here:
> > >
> > > We agree that a combination-specific $w_{\phi, \gamma}(x)$ would indeed be the more expressive choice for learning exact projections onto the appropriate hyperplanes, though it is computationally more costly (note that we have replaced $\theta$ with $\phi$ to emphasize that the parameters of the null space neural network can be distinct from the neural network for the target function). We also agree that learning a single $w_{\phi}(x)$ across changing combinations could be potentially problematic, since even if it may be reasonable to assume that the input space can be partitioned according to which combination is active, the null-space component may be discontinuous at the boundaries of the partitions and may not be learnable with a single continuous $w_{\phi}(x)$.
> > >
> > > Nonetheless, precisely due to the fact that having combination-specific $w_{\phi, \gamma}(x)$ would be too costly, the "trick" we are proposing is to have a single shared $w_\phi(x)$, and to show that this is sufficient for the universal approximation property in Theorem 3.5.
> > >
> > > Specifically, starting at the discussion following Eq. (14) in its proof, there exist target combination-specific null-space components $w_{t, \gamma}(x)$ such that $\lVert w_{t,\gamma}(x)\rVert_p \leq \lVert f_\theta(x) - f_t(x)\rVert_p$, which always exists since 0 satisfies the inequality. Additionally, there exists a single continuous target function $w_{t}(x)$ such that $\lVert w_{t}(x)\rVert_p \leq \lVert w_{t,\gamma}(x)\rVert_p $ for all combinations $\gamma$, which again always exists since 0 satisfies the inequality. Then, from the universal approximation theorem of NNs for the single target null space term as well as the $f_t(x)$ function, there exist a learned null-space vector $w_\phi(x)$ that satisfies $\lVert w_\phi(x) - w_t(x)\rVert_p < K$ and a learned function $f_\theta(x)$ that satisfies $\lVert f_\theta(x) - f_t(x)\rVert_p < K$ with $K = \frac{\epsilon}{3+3\sqrt{n_{out}}}$. Hence, we can bound
> > > $\lVert w_\phi(x)\rVert_p=\lVert w_\phi(x) - w_{t}(x)+w_{t}(x)\rVert_p \le \lVert w_\phi(x) - w_{t}(x)\rVert_p+\lVert w_{t}(x)\rVert_p \le \lVert w_\phi(x) - w_{t}(x)\rVert_p+\lVert w_{t,\gamma}(x)\rVert_p<2K.$
> > >
> > > From this, the bound for $\lVert f_t - \mathcal{P}^{(\gamma)}\rVert_p$ in the proof becomes  $\lVert f_t - \mathcal{P}^{(\gamma)}\rVert_p < (1+3\sqrt{n_{out}})K$, which is the same as the bound we had in the previous proof (after the revision to accommodate arbitrary $p$ norms, as discussed in the first rebuttal), and thus, the remainder of the proof remains the same to obtain $\lVert \mathcal{P}^* - f_t\rVert_p < \epsilon$, which implies that CAffNet preserves the universal approximation property.
> > >
> > > Intuitively, this is enabled by the inherent mechanism that when $f_\theta$ successfully approximates $f_t$, the null space component becomes vanishingly small. Consequently, whether we use a constraint-specific or a single shared null-space component, the universal approximation property of $f_\theta(x)$ is guaranteed, as long as the component can approximate a function bounded within the threshold $K$.
> > >
> > > We plan to provide these details in the proof of Theorem 3.5 in the final revision, should the paper be accepted, and we will also add a remark in Section 3.2 to discuss the choice of a single shared vs combination-specific null space combinations and point to how the single shared one is sufficient based on the updated proof.
> > >
> > > We hope that our response has adequately addressed your concern, and would warmly welcome any revision to the scores you deem appropriate. We deeply appreciate your time, engagement, and thoughtful questions, which have helped strengthen our paper.

---

### Official Review · Reviewer_GcyK · 2026-03-13

**Soundness:** 3
**Presentation:** 3
**Significance:** 3
**Originality:** 3
**Overall Recommendation:** 5
**Confidence:** 3

**Summary:**

CAffNet is a specialised neural network architecture that naturally enforces a set of given input-dependent affine contraints over the output predictions. CAffNet works by augmenting a base model with an additional neural network that can control any degree of freedom left in the output after the constraints are satisfied. With it, CAffNet is able to always satisfy the output constraints while maintaining the universal approximation property.

**Compliance With Llm Reviewing Policy:**

Affirmed.

**Final Justification:**

The paper is good enough to be accepted, provided that the authors expand on the description of the control policy example in Section 4.3, and correct the number of subconstraint sets they need to the tighter polynomial bound they mention in the rebuttal.

**Key Questions For Authors:**

1) How does the performance of CAffNet scale to high-dimensional problems? How do you deal with the combinatorial explosion in the number of constraint subsets?

**Limitations:**

I suggest to add more discussion on applicability of CAffNet to non-toy control problems and/or high-dimensional problems.

**Strengths And Weaknesses:**

_Soundness._ The main projection mechanism of CAffNet (Equation 4) looks sound. The practical consequences of selecting of k out of m subconstraints (Equation 2) remain a bit unclear to me, as the number of subsets may become combinatorially high. The empirical evidence seems to suggest that CAffNet works well in practice, but it is difficult to ascertain whether such performance generalises to high-dimensional problems.

_Presentation._ The paper is professionally written and quite clear, but it requires considerable background in the topic to fully understand it. In particular, the control policy example in Section 4.3 is not introduced in sufficient detail.

There is also a small typo in Figure 4 ("trainning").

_Significance._ CAffNet generalises previous work on HardNet and removes some of the limitations therein. The new method is much more flexible and can produce practical results in important application domains (constrained optimisation, safe control).

_Originality._ CAffNet introduces a novel projection technique to enforce the input-dependent linear constraints without curbing the universal approximation properties of the base model.

---

> ### Author Rebuttal · Authors · 2026-03-31
>
> Thank you for reviewing our paper and for your positive assessment of both its presentation and technical soundness. We also thank you for your insightful comments and questions, which we address below.
>
> >Scalability: Combinatorial explosion
>
> You (and the other reviewers) are absolutely right that there may be practical concerns for selecting $k$ out of $m$ subconstraints, although empirical evidence suggests that it works well in practice. Honestly, that was one of our own main concerns about the limitation of our projection mechanism. Hence, we have continued to work on mitigating this issue since the initial submission and found that we actually do not need that many combinations and the previous number of combinations was merely a conservative upper bound. In particular, instead of the previous total number of combinations of $\sum_{k=1}^{\min(m,n_{out})} \binom{m}{k}$ in Eq. (2), whose complexity is exponential in $m$, i.e., $O(2^m)$, we can simply replace the number of combinations by $
> m + \binom{m}{\min(m,n_{out})}$ with complexity that is polynomial in $m$, specifically $O(m^{\min(m,n_{out})})$. The proof remained nearly identical with the additional observation that by Lemma 3.3, we only need to consider minimal faces (vertices for the pointed polyhedron or hyperplanes for the non-pointed polyhedron) and every individual constraint, instead of all the combinations, which we can easily incorporate in the revised manuscript, should our paper be accepted.
>
> Moreover, with this reduced number of combinations, our experiments have empirically shown that we can increase the number of constraints in the experiment in Section 4.2 by 5 times (i.e., from 11 to 55) without any problems, where the main restriction is our limited GPU memory, which could be increased if needed.
>
> >Scalability: High-dimensional/large-scale problems
>
> Thank you for asking about the scalability of our approach to more complex, high-dimensional problems. We agree that the chosen network and problem sizes may seem small. However, the main reason we chose these sizes was because we wanted to ensure a direct and fairer comparison with existing benchmark problems in HardNet (Min \& Azizan, 2025) and DC3 (Donti et al., 2021), and these were the network sizes (3 layers, 200 neurons) and environments chosen in those papers.
>
> Nonetheless, we hope to consider more complex real-world experiments in our future work, potentially including hardware demonstrations. As a start, we performed additional experiments to examine the performance of our proposed CAffNet approach with a larger network size, and found that we could increase the number of layers and neurons to 256 and 896, respectively, without any problems, with the limitation again being our limited GPU memory.
>
> >Inadequate details for control policy example
>
> Thank you for pointing out this concern. We apologize for leaving out some details, primarily due to concerns about space. We will be glad to include more details on the control policy example (that are similar to that in HardNet (Min \& Azizan, 2025)) in the final manuscript, should the paper be accepted. Specifically, we plan to explain more about the obstacle avoidance modeling, the smoothing function, and the control barrier functions. If needed, we are open to suggestions on what other details may be helpful, and we will include them in the final version.
>
> >Typo in the legend of Fig. 4
>
> Thank you for catching and pointing out the typo. We will fix it in the revised manuscript.
>
> We hope that our responses have adequately addressed your concerns, and would warmly welcome any revision to the scores you deem appropriate. We sincerely appreciate the thorough and constructive feedback on our paper.

---

> > ### Author Rebuttal · Reviewer_GcyK · 2026-04-01
> >
> > Thanks for the detailed response, I appreciate it.
> >
> > Could you clarify whether your statement "increase the number of layers and neurons to 256 and 896" contains a typo, or you actually tried running networks with 256 layers?

---

> > > ### Author Response · Authors · 2026-04-07
> > >
> > > We are glad to hear that most of your concerns have been addressed, and thank you for the follow-up clarification question. This was actually not a typo (we did try running networks with 256 layers), although we do realize that it is somewhat unconventional/"absurd" to try running it with so many layers. We were merely trying to see how far we could push our approach to probe its limits, as part of our attempt to allay another reviewer's concern that the neural networks in our experiments (3x200 neurons) were potentially too small/unrealistic. We would also like to clarify that the large number of layers in the neural networks was not necessary for the benchmark experiments in our paper, for which the 3x200 networks were already sufficient. The larger network size was mainly for stress testing our approach and will not be used in those experiments in our final revision, should the paper be accepted.
> > >
> > > We hope that our response has adequately addressed your concern, and would warmly welcome any revision to the scores you deem appropriate. We deeply appreciate your time, engagement, and thoughtful questions, which have helped strengthen our paper.

---

### Decision · Program_Chairs · 2026-04-30

**Decision:**

Accept (regular)

**Comment:**

This paper proposes CAffNet, which is a specialised neural network architecture designed to inherently enforce a set of input-dependent affine constraints on output predictions. It operates by augmenting a base model with an auxiliary neural network that governs any remaining degrees of freedom in the output once the constraints have been satisfied. Through this design, CAffNet guarantees constraint satisfaction at the output while preserving the universal approximation property.

The overall consensus that this is a solid paper with good contributions. The authors have also done a great job during the rebuttal to convince the reviewers about the merits of their work. Given this, I strongly support the paper's acceptance.